# Arbitrary linear transformations for photons in the frequency synthetic dimension

Siddharth Buddhiraju [1], Avik Dutt [1], Momchil Minkov [1], Ian A. D. Williamson [1] & Shanhui Fan [1✉]

Arbitrary linear transformations are of crucial importance in a plethora of photonic applications spanning classical signal processing, communication systems, quantum information processing and machine learning. Here, we present a photonic architecture to achieve arbitrary linear transformations by harnessing the synthetic frequency dimension of photons. Our structure consists of dynamically modulated micro-ring resonators that implement tunable couplings between multiple frequency modes carried by a single waveguide. By inverse design of these short- and long-range couplings using automatic differentiation, we realize arbitrary scattering matrices in synthetic space between the input and output frequency modes with near-unity fidelity and favorable scaling. We show that the same physical structure can be reconfigured to implement a wide variety of manipulations including single-frequency conversion, nonreciprocal frequency translations, and unitary as well as non-unitary transformations. Our approach enables compact, scalable and reconfigurable integrated photonic architectures to achieve arbitrary linear transformations in both the classical and quantum domains using current state-of-the-art technology.

[1] Ginzton Laboratory, Department of Electrical Engineering, Stanford University, Stanford, CA, USA. ✉email: shanhui@stanford.edu

Arbitrary linear transformations in photonics[1–3] are of central importance for optical quantum computing[4], classical signal processing and deep learning[5–10]. A variety of architectures are being actively studied to implement linear transformations for quantum computation and photonic neural networks, including those based on Mach–Zender interferometers (MZI)[4,5], microring weight banks[6,7,11], phase-change materials[8,9], and diffractive metasurfaces[10]. All such approaches use path encoding of photons in real space. By contrast, implementing such linear transformations in the frequency space would open avenues beyond those possible with previously reported architectures, which are inherently time-invariant. For example, frequency-space transformations allow spectrotemporal shaping of light and generation of new frequencies, with wide-ranging applications in frequency metrology, spectroscopy, communication networks, classical signal processing[12–14] and linear optical quantum information processing[15–24]. Nonlinear optics has traditionally been the workhorse for such spectrotemporal shaping, but the requirement of high-power fields and the difficulty of implementing arbitrary linear transformations motivates new architectures for manipulating states in the frequency domain. To that end, photonic synthetic dimensions offer an attractive solution to implement linear transformations in a single physical waveguide by harnessing the internal degrees of freedom of a photon[25–32]. Synthetic frequency dimensions in particular offer a small spatial footprint and inherent reconfigurability since multiple frequency modes can be addressed simultaneously, and the short- and long-range coupling[29,30,33,34] between them can be controlled by applying an appropriate time-domain signal to a modulator.

Previous works have considered implementing photonic linear transformations using different frequency channels in parallel but without frequency conversions among them[6,7,9,11] by demultiplexing the different frequencies into separate spatial channels. Additionally, optimized fast modulation has been used for tailoring single photon spectra from two-level quantum emitters[35], or for quantum frequency conversion[15] and linear optical quantum computation[17,36], where the modulator is used as a generalized beam splitter in synthetic frequency dimensions. However, the design of an entire scattering matrix that implements an arbitrary $N \times N$ linear transformation in synthetic space, which is essential for many applications in quantum information processing and neural networks, has not yet been shown.

Here, we show that arbitrary linear transformations can be performed directly in the synthetic space spanned by the different frequency modes carried by a single physical waveguide. We use gradient-based inverse design to automate the process of designing the linear transformations, and demonstrate that a wide variety of transformations can be realized. As examples, we show single-frequency conversion, nonreciprocal frequency translations as well as general arbitrary unitary and non-unitary transformations, all achieved with high fidelities in a fully reconfigurable fashion.

## Results

**Theory.** Consider a ring of radius $R$ formed by a single mode waveguide with a refractive index $n$. The ring is coupled to an external waveguide of the same refractive index. Assuming sufficiently weak coupling between the ring and the external waveguide and neglecting group-velocity dispersion, the eigenmodes of the ring occur at frequencies $\omega_m = \omega_0 + m\Omega_R$, where $\omega_0$ is the central frequency, $m$ is an integer and $\Omega_R = c/nR$ is the free spectral range (FSR) of the ring in angular frequency units, with $c$ being the speed of light in vacuum. These eigenmodes take the form $e^{-i(m_0+m)\phi}$, where $m_0$ denotes the angular momentum of the

$0^{\text{th}}$ mode and $\phi$ is the azimuthal coordinate of the ring. Corresponding to these eigenmodes, we define $a_m(t)e^{i\omega_m t}$ to be the amplitude of the mode centered at $\omega_m$, normalized such that $|a_m(t)|^2$ corresponds to the photon number in the $m^{\text{th}}$ mode. Likewise, we define $s_m^{\pm}(t)e^{i\omega_m t}$ to be the amplitudes of the modes of the external waveguide at the input and output ports, respectively, as shown in Fig. 1b. The coupling between the ring modes and waveguide modes at frequency $\omega_m$ is described by an external coupling rate $\gamma_m^e$, while other losses occurring in the ring, such as absorption or bending loss, are captured by an internal decay rate $\gamma_m^i$. Lastly, we assume that the dielectric constant of the ring is modulated using an electro-optic modulator in the form $\sum_{l=1}^{N_f} \delta\epsilon_l(\phi)\cos(l\Omega_R t + \theta_l)$, where $\delta\epsilon_l$ is the depth of the modulation and $\theta_l$ is the phase of the modulation at frequency $l\Omega_R$. The angular dependence $\delta\epsilon_l(\phi)$ occurs due to the physical localization of the electro-optic modulator to a specific range of $\phi$, as shown in Fig. 1. The dynamics of the coupled ring-waveguide system can be described by a coupled-mode theory (see Supplementary Note 1) given by:

$$-id_t a_m = i(\gamma_m^e + \gamma_m^i)a_m + \sqrt{2\gamma_m^e}s_m^+ \\ + \sum_{l=1}^{N_f}(\kappa_l a_{m-l} + \kappa_{-l}a_{m+l}), \quad (1)$$

$$s_m^- = s_m^+ + i\sqrt{2\gamma_m^e}a_m, \quad (2)$$

where

$$\kappa_{\pm l} = -\frac{\alpha_l}{4n^2}e^{\mp i\theta_l}\int_0^{2\pi}e^{\mp il\phi}\delta\epsilon_l(\phi)d\phi \quad (3)$$

is the modulation-induced coupling between the modes of the ring, with $\alpha_l$ describing the radial and zenith-angle overlap of the eigenmodes of the ring with the electro-optic modulator (see Supplementary Note 1).

If the $\delta\epsilon_l$'s are real, i.e., only the real part of the refractive-index is modulated, then $\kappa_l^* = \kappa_{-l}$. Therefore, the modulation conserves the total photon number summed across all frequency channels. Further, if $\gamma_m^i$ are negligible, then no photons are lost to absorption or radiation. Under these conditions, the setup of Eqs. (1–2) implements a unitary transformation between the fields $s_m^+$ at the input ports and the fields $s_m^-$ at the output ports. This unitary transformation can be obtained by first converting Eq. (1) to the frequency domain, resulting in

$$\mathbf{a} = [\Delta\omega - i\Gamma - \mathcal{K}]^{-1}\sqrt{2\Gamma}\mathbf{s}^+, \quad (4)$$

where $\mathbf{a} = \{\dots a_{-1}, a_0, a_1, \dots\}^t$, $\mathbf{s}^{\pm} = \{\dots s_{-1}^{\pm}, s_0^{\pm}, s_1^{\pm}, \dots\}^t$, $\Gamma = \text{diag}(\dots \gamma_{-1}^e, \gamma_0^e, \gamma_1^e, \dots)$, $\Delta\omega$ is a constant detuning of the equally spaced frequencies of input comb $\mathbf{s}^+$ from the ring's resonant frequencies, and $\mathcal{K}_{mm'} \equiv \kappa_{m-m'}$ as defined by Eq. (3). Then, from Eq. (2), we obtain $\mathbf{s}^- = \mathcal{M}\mathbf{s}^+$, where

$$\mathcal{M} = \left[\mathcal{I} + i\sqrt{2\Gamma}[\Delta\omega - i\Gamma - \mathcal{K}]^{-1}\sqrt{2\Gamma}\right]. \quad (5)$$

A direct verification of the unitarity of $\mathcal{M}$ is included in Supplementary Note 2. In the idealized situation as described above, where the ring-waveguide system is assumed to be single-moded over a broad bandwidth and is free from group velocity dispersion, the matrix $\mathcal{M}$ is infinite-dimensional. In practice, the dimensionality of the scattering matrix can be controlled by introducing a "truncation" along the frequency dimension. Such a truncation can be implemented using one or more auxiliary rings coupled to the main ring (see Supplementary Note 3). The auxiliary rings couple to and perturb a few modes immediately outside the $(2N_{sb} + 1)$ modes around the $0^{\text{th}}$ mode, dispersively shifting and splitting them. These perturbed modes have

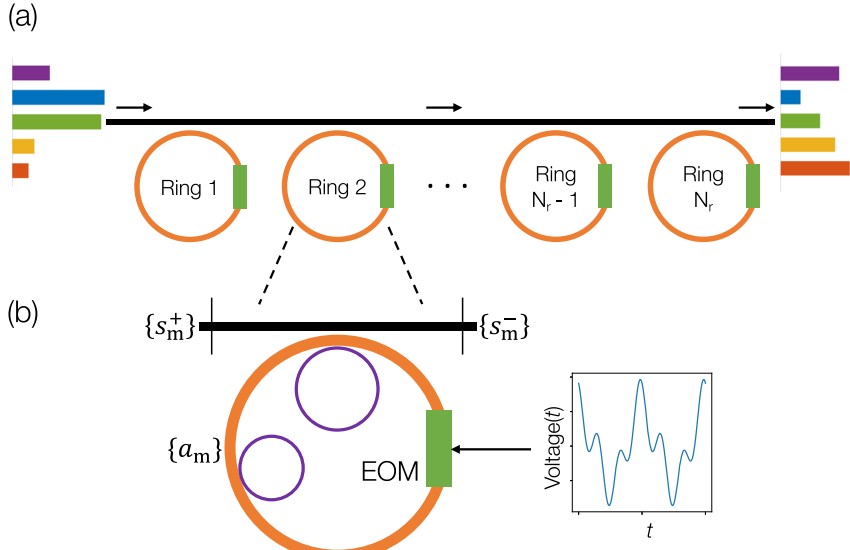

**Fig. 1 Setup to implement arbitrary linear transformations in frequency space. a** An array of dynamically modulated rings (orange) coupled to an external waveguide. The green blocks represent electro-optic modulators (EOMs) and the black line is an external waveguide coupling to each of the rings. The output spectrum on the right is the result of the transformation implemented by the system on the input spectrum (left). **b** Detailed view of a single ring depicting the waveguide port inputs and outputs $s_m^\pm$. The smaller purple circles indicate auxiliary rings that couple selectively to modes $a_m$ of the larger orange ring to implement frequency-dimension truncation to the ring spectrum. The time-periodic voltage profile applied to the EOM is a result of the inverse-design algorithm.

frequencies such that the modulation tones of $l\Omega_R$ cannot couple these modes to the $(2N_{sb} + 1)$ modes of interest. Therefore, the total number of modes under consideration in the coupled ring-waveguide system is $2N_{sb} + 1$, and the scattering matrix defined in Eq. (5) is of size $(2N_{sb} + 1) \times (2N_{sb} + 1)$.

The main objective of our paper is to show that an arbitrary scattering matrix of size $(2N_{sb} + 1) \times (2N_{sb} + 1)$ can be created. To that end, we first note that the number of real degrees of freedom in the scattering matrix (Eq. (5)) of a single ring under modulation is equal to twice the number of distinct modulation tones, $2N_f$, provided the modulation amplitudes $\delta\epsilon_l$ and phases $\theta_l$ are independently controllable. Since the system is truncated to have $2N_{sb} + 1$ frequencies, the largest harmonic of $\Omega_R$ that will result in nonzero coupling between any two modes is $2N_{sb}$, i.e., $N_f \leq 2N_{sb}$. Since an arbitrary unitary matrix of size $(2N_{sb} + 1) \times (2N_{sb} + 1)$ has $(2N_{sb} + 1)^2$ real degrees of freedom whereas $N_f \leq 2N_{sb}$, we conclude that a single modulated ring is insufficient to approximate an arbitrary unitary matrix to a high degree of accuracy, even if all modulation tones up to $2N_{sb}\Omega_R$ are used. To overcome this problem, notice that products of unitary transformations are also unitary[37]. Therefore, as shown in Fig. 1a, instead of a single ring, we consider a sequence of $N_r$ number of rings with each ring providing $N_f$ complex degrees of freedom. Thus, if the total degrees of freedom in series of rings coupled to the waveguide, given by $2N_fN_r$, exceeds $(2N_{sb} + 1)^2$, then the setup of Fig. 1a should be able to approximate an arbitrary unitary transformation to a high degree of accuracy.

Below, we optimize these $2N_fN_r$ degrees of freedom to enable physical approximation of arbitrary unitary and certain non-unitary transformations. For unitary transformations or parts thereof, we use as the objective function the fidelity, which measures the accuracy of an approximation $V$ to a unitary transformation $U$:

$$\mathcal{F}(U, V) = \frac{|\langle U, V \rangle|}{\sqrt{||U||_F ||V||_F}}, \qquad (6)$$

where $\langle U, V \rangle = \sum_{ij} U_{ij}^* V_{ij}$ is the element-wise inner product and

$||U||_F = \sqrt{\sum_{ij}|U_{ij}|^2}$ is the Frobenius norm. The use of an absolute value in Eq. (6) allows for the tolerance of a single global phase, i.e., if $\mathcal{F}(U, V) = 1$, then the transformation $V$ achieved by the architecture is equal to $Ue^{i\Phi}$ for some phase $\Phi$. To achieve a high fidelity for a given target matrix we use gradient-based inverse design to optimize the parameters of the modulated system. To enable such optimization, we implemented a numerical model of the unitary transformations defined by Eq. (5) in an automatic differentiation framework[38]. While explicitly defined adjoint variable methods have been widely used for photonic inverse design[39], automatic differentiation is the generalization of the adjoint variable methods to arbitrary computational graphs. Automatic differentiation has recently been successfully applied to the inverse design of photonic band structures[40] as well as photonic neural networks[41], where explicit adjoint methods are challenging to implement. Here, automatic differentiation enables the efficient computation of the gradients of a scalar objective function with respect to complex control parameters, which in this case are the coupling constants $\kappa_{\pm l}$ as defined in Eq. (3). The advantage of using automatic differentiation is that one needs only to implement the computational model as described above, while the automatic differentiation framework manages the gradient computation through an efficient reverse-mode differentiation. Using the gradients from automatic differentiation, the Limited-memory Broyden–Fletcher–Goldfarb–Shanno (LBFGS) algorithm[42] is used for optimization.

**Implementation of linear transformations.** For the results in this Section, we assume that the ring-waveguide system under consideration operates with $N_{sb} = 2$, i.e., 5 equally spaced lines followed by at least 4 perturbed lines on each side. The five relevant modes are indexed $\{-2, -1, 0, 1, 2\}$. For simplicity, we assume that all five ring modes couple to the waveguide with equal strength, i.e., $\gamma_m^e \equiv \gamma$ and $\gamma_m^i = 0 \,\forall m$. We also assume that the source frequencies in the waveguide are on resonance with the ring, i.e., $\Delta\omega = 0$ in Eq. (5). Examples of finite intrinsic loss ($\gamma_m^i \neq 0$) and non-uniform detuning ($\Delta\omega \neq 0$) are considered in

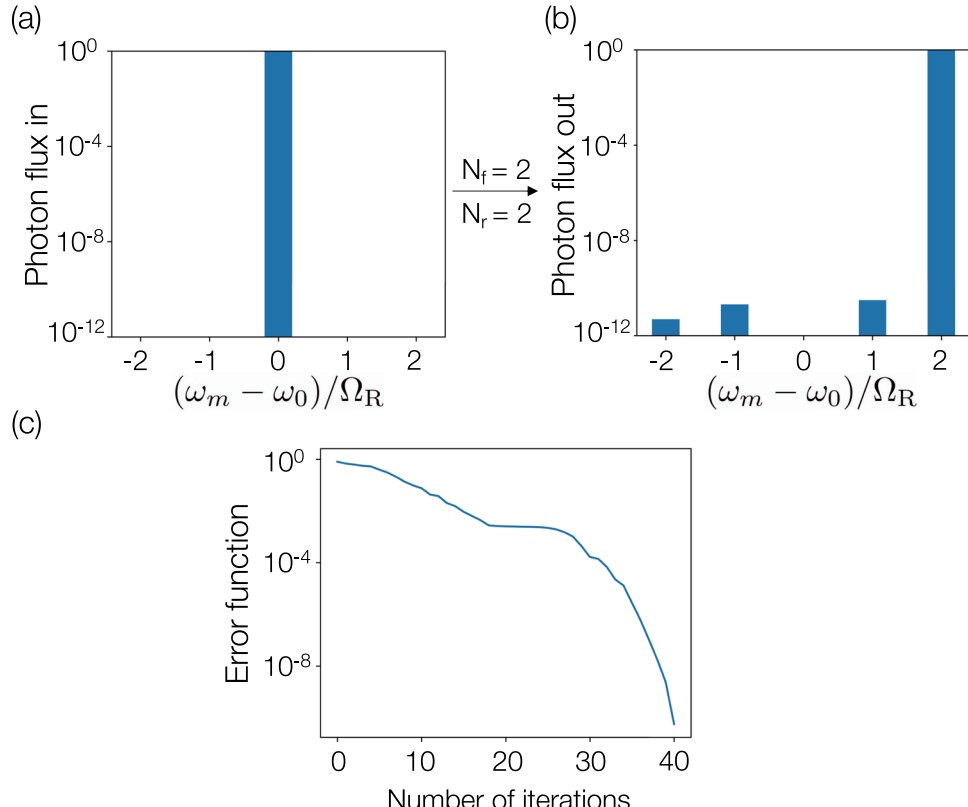

**Fig. 2 Frequency translation.** A two-ring system with two modulation tones per ring ($N_r = 2$, $N_f = 2$) demonstrating a conversion efficiency exceeding $1 - 10^{-5}$ from mode 0 to +2. The bar plots show log-scale photon flux in each mode at **a** the input, and **b** the output. **c** The error function shown as a function of the number of iterations of the optimization algorithm to achieve the conversion efficiency of **b**. Modulation parameters are provided in Supplementary Note 6.

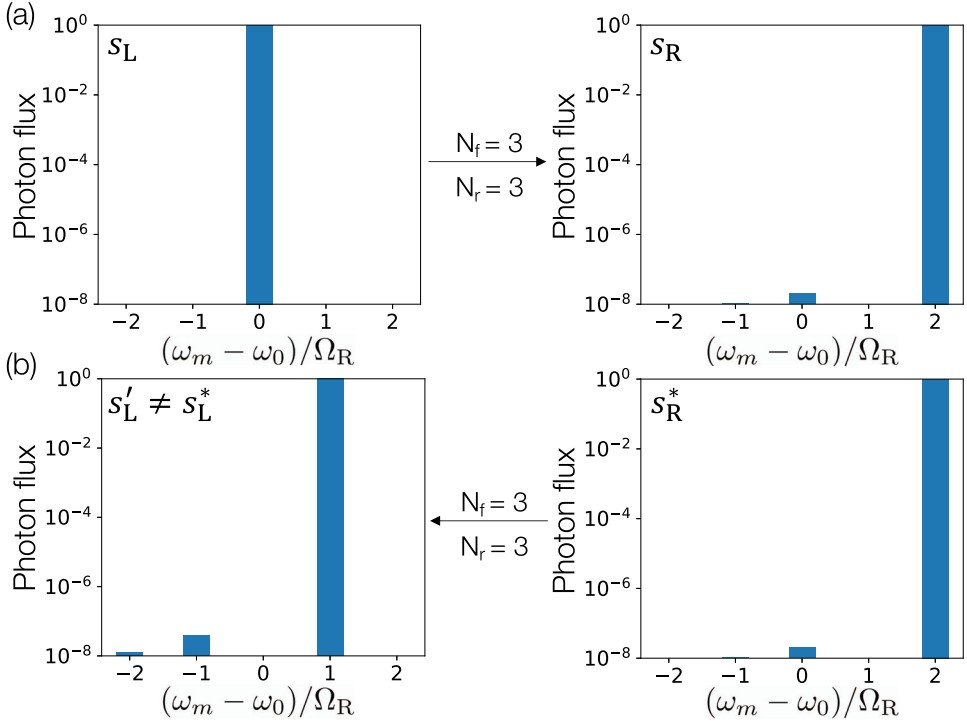

**Fig. 3 Nonreciprocal frequency conversion.** A three-ring system with three modulation tones per ring, demonstrating **a** a conversion efficiency exceeding $1 - 10^{-5}$ from mode 0 to mode +2 in forward propagation. The input and output field profiles are indicated by $s_L$ and $s_R$, respectively, and **b** the complex-conjugated output profile, $s_R^*$, injected back into the output port results in a conversion efficiency exceeding $1 - 10^{-5}$ from mode +2 to +1 instead of mode 0 in backward propagation through the same system, indicating highly efficient nonreciprocal frequency shifts. Modulation parameters are provided in Supplementary Note 6.

Supplementary Notes 4 and 5. Note that the different source frequencies' phases should not drift with respect to each other during the timescale of the transformation. To ensure such phase coherence between the different input frequency modes, the source could be a mode-locked laser or an electro-optic frequency comb with a tailored amplitude/phase spectrum to implement the input vector. Alternatively, active phase stabilization could be implemented to compensate for slow-timescale phase drifts. Under the assumptions made in this Section, the transformation in Eq. (5) is completely determined by the ratios $\kappa_l/\gamma$, where $\kappa_l$ is controlled by the index perturbation amplitude $\delta\epsilon_l$ and phase $\theta_l$, as described by Eq. (3). Therefore, we optimize the amplitude and phase of $\kappa_l$ (in units of $\gamma$) for $N_r$ rings and $N_f$ modulation tones per ring to implement a variety of transformations. Note that since we only optimize for the ratios $\kappa_l/\gamma$, our approach is robust to variations in $\gamma$ during fabrication.

First, we consider the application of such ring-waveguide networks to implement high-fidelity frequency translation that is useful for frequency-domain beam-splitters or single-qubit gates. As an example, we show a design where an input signal in mode 0, after forward propagation through the network, results in a complete conversion to mode $+2$. Using our inverse-design framework, such a frequency translation corresponds to designing only one column of a unitary transformation and can be achieved

with a fidelity exceeding $1 - 10^{-5}$ using just two rings and two modulation tones per ring, as shown in Fig. 2a, b. In Fig. 2c, we present the error function versus the number of iterations. The error function is defined as $1 - F_{+2}$, where $F_{+2}$ is the normalized output photon flux in the mode $+2$. After a few iterations, almost all the photon flux is converted to frequency $\omega_{+2}$ at the output.

In addition to such high-fidelity frequency conversion implemented in forward propagation through the network, the transformations achieved in this architecture can be different in forward and reverse propagation due to the relative phase shift between the modulation tones across the different rings and the explicit time-varying nature of the dynamically modulated system[43]. This is in sharp contrast with MZI-based architectures, which are inherently reciprocal. As an example, we show in Fig. 3 that we can simultaneously realize with a fidelity exceeding $1 - 10^{-5}$ a frequency shift, say, $0 \rightarrow 2$, in forward propagation (Fig. 3a) and a different shift, say, $2 \rightarrow 1$, in reverse propagation (Fig. 3b) with three modulated rings.

Achieving frequency shifts using modulated rings, as shown in Figs. 2 and 3, requires designing only one and two columns of the $5 \times 5$ unitary matrix, respectively. On the other hand, if the number of modulation tones $N_f$ and/or the number of rings $N_r$ are increased, an arbitrary unitary transformation can be achieved with a high fidelity. As an example, we depict in Fig. 4a a $5 \times 5$

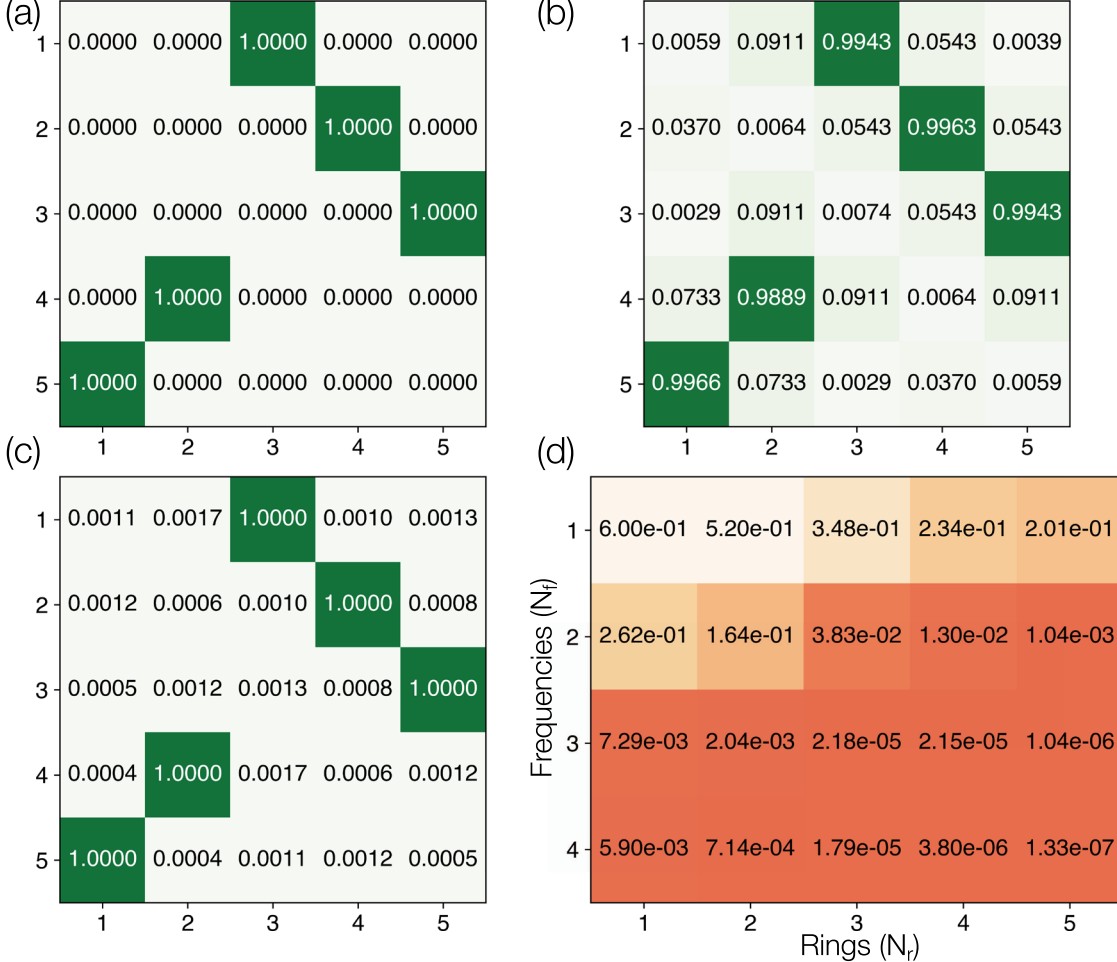

**Fig. 4 Implementing elementwise amplitudes of a matrix. a** A $5 \times 5$ permutation matrix to be implemented by the ring-waveguide system. The amplitudes of the matrix elements are indicated along with a green colormap. Element-wise amplitudes of the optimized result using four modulation tones ($N_f = 4$) and **b** one ring ($N_r = 1$), achieving a fidelity of $1 - 5.9 \times 10^{-3}$, and **c** four rings ($N_r = 4$), achieving a fidelity of $1 - 3.8 \times 10^{-6}$. **d** One minus the maximum fidelities achieved by the inverse-design algorithm as a function of $N_r$ and $N_f$. A value closer to zero indicates a better performance. Modulation parameters for **c** are provided in Supplementary Note 6.

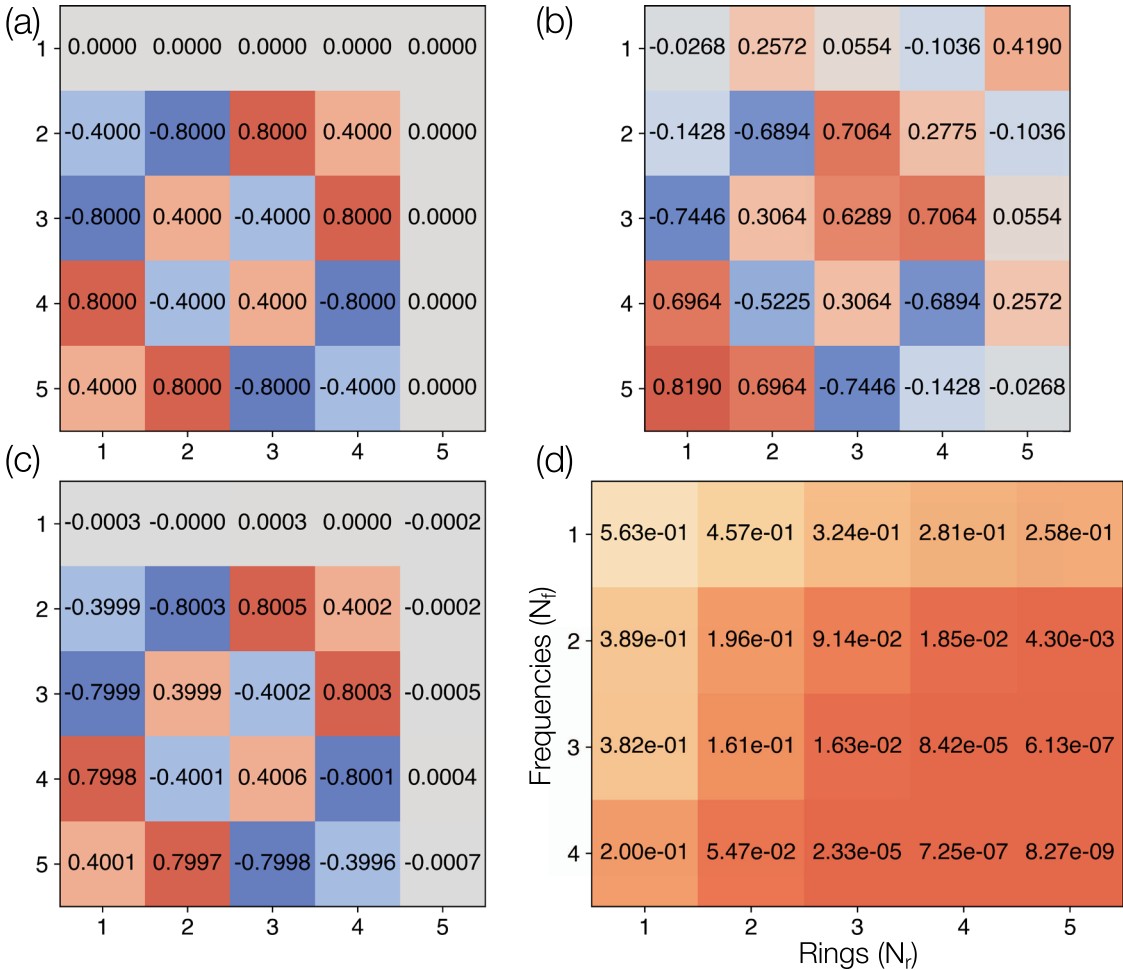

**Fig. 5 Implementing elementwise phases of a matrix. a** Element-wise phase as a fraction of $\pi$ of the $5 \times 5$ Vandermonde matrix implementing the discrete Fourier transform. Element-wise phase achieved by the inverse-design algorithm for **b** $N_r = 1$ and $N_f = 4$, with a fidelity of 0.8 and global phase of $0.099\pi$ and **c** $N_r = 4$ and $N_f = 4$, with a fidelity of $1 - 7.25 \times 10^{-7}$ and global phase $0.596\pi$. **d** One minus the maximum fidelities achieved by the inverse-design algorithm as a function of $N_r$ and $N_f$. A value closer to zero indicates a better performance. Modulation parameters for **c** are provided in Supplementary Note 6.

permutation matrix $U$, defined by $U_{13} = U_{24} = U_{35} = U_{42} = U_{51} = 1$, and zero otherwise. In Fig. 4b, we present the amplitudes of the matrix achieved using one ring and four modulation tones, resulting in a fidelity of $1 - 5.9 \times 10^{-3}$. With four rings and four modulation tones, the fidelity is boosted to over $1 - 3.8 \times 10^{-6}$, as shown by the amplitudes in Fig. 4c. In Fig. 4d, we tabulate as a function of $N_r$ and $N_f$ one minus the maximum fidelities obtained in approximating the $5 \times 5$ permutation matrix, showing that very high fidelities can be achieved using a wide variety of $N_r$ and $N_f$ combinations.

In Fig. 4, we considered only the accuracy of the amplitudes achieved by our inverse-design approach. We now show that our architecture can also capture the phase of an arbitrary unitary transformation with a high fidelity. To demonstrate this, we consider a normalized $5 \times 5$ Vandermonde matrix, which is used to implement the discrete Fourier transform. This unitary transformation, defined by $U_{mn} = e^{-2\pi imn/5}/\sqrt{5}$, has a constant amplitude across its matrix elements but significantly varying phase, as shown in Fig. 5a. With the use of one ring and four modulation tones, the inverse-design algorithm is able to achieve a fidelity of 0.8, with the corresponding phase profile shown in Fig. 5b up to a global phase of $0.0099\pi$. As depicted in Fig. 5c, a significantly better performance is possible with the use of four

rings and four modulation tones per ring, achieving a fidelity of $1 - 7.25 \times 10^{-7}$ with a global phase of $0.596\pi$. A map of one minus the maximum fidelities achieved by our inverse design approach as a function of the number of rings and modulation tones is shown in Fig. 5d.

While unitary transformations are usually required for quantum information processing, matrices used in classical signal processing and in neural networks are in general non-unitary. The architecture presented thus far can also be used to implement non-unitary matrices with singular values less than or equal to one using one of two techniques. First, such non-unitary matrices can provably be embedded in larger unitary matrices[44] using their singular value decomposition. Subsequently, the larger unitaries can be implemented using refractive index modulation as discussed thus far. As an example, we consider the following $3 \times 3$ non-unitary matrix that was randomly generated subject to the constraint that its largest singular value is equal to one:

$$M = \begin{pmatrix} 0.4993e^{i0.2483\pi} & 0.3135e^{i0.3251\pi} & 0.3150e^{i0.1519\pi} \\ 0.2580e^{i0.4129\pi} & 0.2888e^{i0.1608\pi} & 0.4420e^{i0.0492\pi} \\ 0.5277e^{i0.2319\pi} & 0.2382e^{i0.1053\pi} & 0.1992e^{i0.4087\pi} \end{pmatrix}. \quad (7)$$

The singular values of $M$ are 1, 0.3755 and 0.1421, respectively.

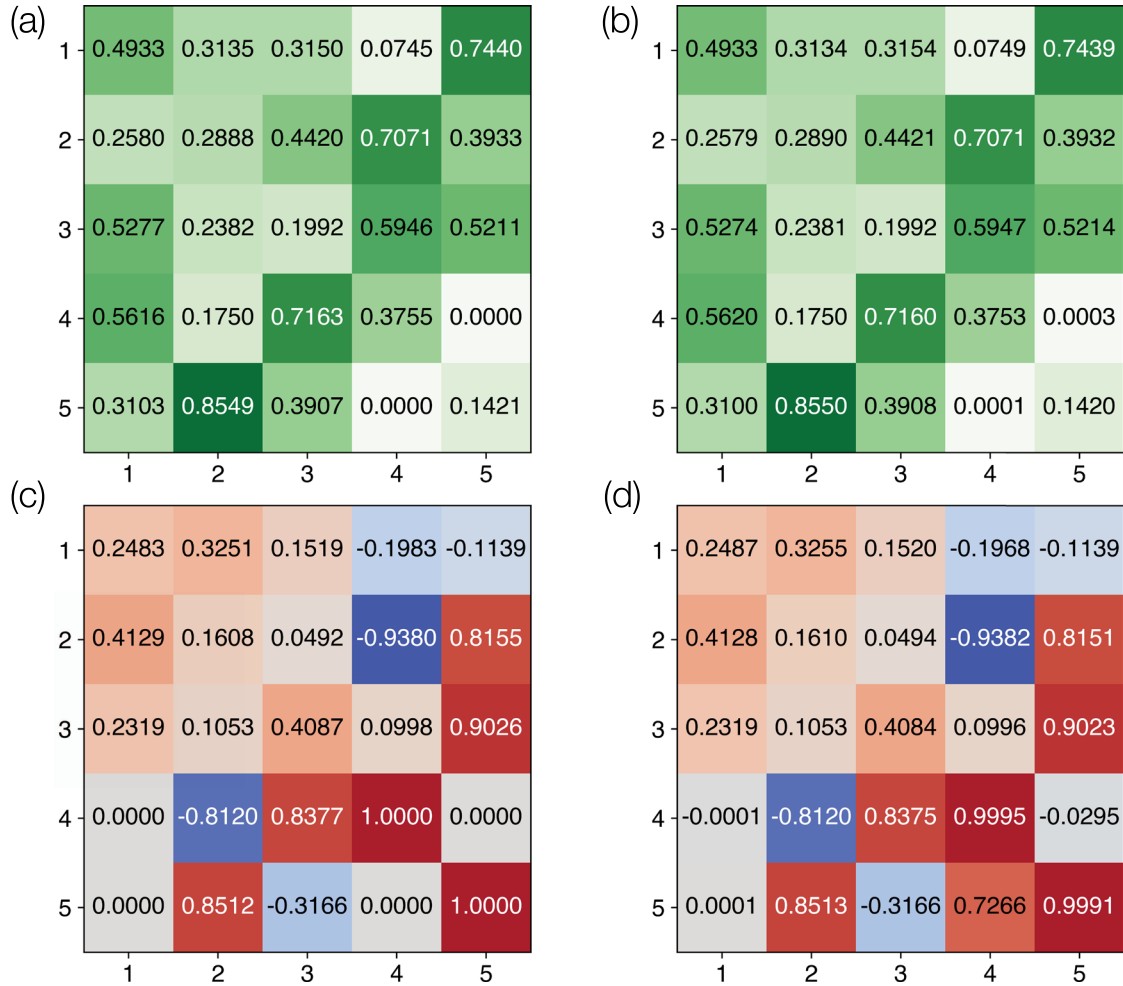

**Fig. 6 Implementing non-unitary transformations with unitary embedding.** The target non-unitary matrix, located in the upper-left $3 \times 3$ section of the matrix, is first extended to a unitary $5 \times 5$ target matrix. The element-wise **a** target amplitude, **b** achieved amplitude, **c** target phase, and **d** achieved phase as a fraction of $\pi$ are shown. A near ideal implementation was achieved using $N_r = 4$ and $N_f = 4$ with a fidelity exceeding $1 - 10^{-5}$. Modulation parameters are provided in Supplementary Note 6.

Since there are two singular values less than 1, $M$ can be extended into a unitary matrix by adding two dimensions. The element-wise amplitude and phase corresponding to the extended $5 \times 5$ unitary matrix are shown in Fig. 6a, c, respectively. Using four rings ($N_r = 4$) and four modulation tones per ring ($N_f = 4$), our inverse-design algorithm achieves the extended unitary matrix with a fidelity exceeding $1 - 10^{-5}$, as shown in Fig. 6b, d. Notice that the phase of element (5, 4) is significantly different between Fig. 6c, d, but this is because the target amplitude for this element is zero. As an alternative approach, amplitude modulation, where the imaginary part of the refractive index is also modulated, can also be used to directly implement non-unitary matrices since the transformation of Eq. (5) is non-unitary under modulation of the imaginary part of the refractive index. Lastly, in order to implement matrices with singular values greater than 1, a gain element is necessary. For such matrices, a scaled version such that the singular values are below 1 can first be implemented using the methods outlined above, after which a uniform amplification for all frequency channels can rescale the matrix to its intended form.

## Discussion

We have shown that combining the concepts of synthetic dimensions and inverse design enables the implementation of versatile linear transformations in photonics. A major advantage

of using synthetic frequency dimensions for implementing an $N \times N$ linear transformation is that only O($N$) photonic elements (modulators in our case) need to be electrically controlled. This is in contrast to real-space dimensions using path-encoding, such as MZI meshes or crossbar arrays, where the full O($N^2$) degrees of freedom need to be electrically controlled. Such control is non-trivial both from a scalability perspective as well as from a practical geometrical perspective of connecting $N^2$ tunable elements (e.g. phase-shifters) to their driving electronics off-chip. The reduction in the number of individually controlled elements from O($N^2$) to O($N$) in our scheme comes from the fact that the driving signal on each of the $N_r$ EOMs can simultaneously address $N_f$ frequency modes in the synthetic dimension.

Future work could leverage synthetic frequency dimensions for complicated quantum information protocols beyond single-qudit unitary transformations, such as realizing probabilistic entangling gates for linear optical quantum computing (LOQC)[17,36]. In particular, spectral LOQC using EOMs and pulse shapers has been shown to be universal for quantum computation[17]. However, pulse shapers involve demultiplexing the frequency modes into distinct spatial channels using gratings to apply mode-by-mode phase shifts, and limit the number of modes that can be accommodated within the modulator bandwidth due to a finite spectral resolution, thus reducing the benefit of using synthetic

frequency dimensions. Such pulse shapers are also lossy and challenging to integrate on chip. Our architecture obviates the pulse shaper by exclusively using EOMs. The advent of ultralow-loss nanophotonic EOMs in lithium niobate[45,46], as well as progress in silicon[47,48] and aluminum nitride[49] makes our architecture fully compatible with on-chip integration, since modulation at frequencies exceeding the ring's FSR have been demonstrated[14,47,50].

For applications in neural networks, the performance of our architecture in terms of the speed, compute density and energy consumption for multiply-and-accumulate (MAC) operations is important[51]. Assuming we need $N$ modulation tones and $N$ rings with FSR $\Delta f = \Omega_R/2\pi$ to implement a matrix, we can input information encoded in the $N$ frequencies and read out the matrix-vector product, which amounts to $N^2$ MAC operations. Since we need a frequency-resolved measurement, the fastest readout bandwidth is $\Delta f$. We assume that the input data can be prepared at speed comparable to or faster than the readout speed. Then, the computational speed in MACs per second is given by

$$C = N^2 \Delta f. \tag{8}$$

The maximum number of channels is limited by the FSR and the modulation bandwidth. If we utilize the whole available bandwidth, $B = N\Delta f$, then the speed is

$$C = NB. \tag{9}$$

For a modulation bandwidth of 100 GHz and an FSR of 100 MHz (such that $N = 1000$), this yields a speed $C = 10^{14}$ MACs per second or 100 TMAC per second, which is comparable with MZI meshes[5,51,52]. Although achieving such small FSRs on chip is challenging, recent progress in integrating low-loss delay lines on chip[53,54] holds promise, since meter-scale delays were reported in an 8 mm$^2$ footprint using spiral resonators, corresponding to an equivalent FSR of ~350 MHz[53]. These design techniques can be extended to lithium niobate rings with high modulation bandwidths[14,46].

To optimize for computation density, i.e. MACs per second per unit area[51], one can use a larger FSR $\Delta f = 1$ GHz, in a 1-mm$^2$ footprint, and combine synthetic frequency dimensions within each 100-GHz modulation bandwidth with wavelength-division multiplexed channels separated by 100-GHz-wide stopbands, to parallelize several uncoupled MAC operations across the 5 THz telecommunications band, as has been done for crossbar arrays[6,7,9,51]. This leads to a compute density of ~10 TMAC s$^{-1}$ mm$^{-2}$, which is much better than MZI meshes and comparable with standard silicon microring crossbar arrays[51], with the added advantage of only O($N$) electronically controlled elements. We anticipate that future progress in modulation speed and power using high-confinement integrated photonic platforms will push these current estimates further, leading to experimental implementations of MAC operations using the architecture proposed here with improvements in complexity, speed, power and footprint.

## Data availability
The data related to this study is available in the manuscript and the Supplementary Materials. Additional data is available from the authors upon reasonable request. Correspondence and requests for materials should be addressed to S.F.

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

## Acknowledgements
This work is supported by the U.S. Air Force Office of Scientific Research (FA9550-17-1-0002, FA9550-18-1-0379). S.B. acknowledges the support of a Stanford Graduate Fellowship.

## Author contributions
S.B., M.M., and I.A.D.W. conceived the project. S.B., A.D., and M.M. performed the research. S.F. supervised the research. All authors analyzed the results and contributed to discussions. S.B., A.D. and S.F. wrote the manuscript.

## Competing interests
The authors declare no competing interests.
