## [Peer Review File · Nature Communications]

REVIEWER COMMENTS

Reviewer #2 (Remarks to the Author):

This is a theoretical proposal of a scheme to implement linear transformations of frequency combs. This is based on a chain of ring resonators, each driven by a modulator with specially optimized driving function. The potential to realize such arbitrary unitary and non-unitary transformation would be of interest for multiple applications in classical and quantum photonics, and thus can stimulate further experimental works. Therefore, this manuscript appears timely and suitable for a readership of Nature Communications.

The manuscript is clearly written. I have the following comments and suggestions:

1) While the intrinsic resonator loss is included in Eq. (1), it is assumed to be zero in the rest of the manuscript. In presence of such loss, would it be possible to still implement the unitary matrix transformations, up to an overall loss (such that the largest singular value is less than one)? A numerical example, e.g. in Supplementary materials, would be very useful.

2) The model assumes that the frequency detuning of all the rings in a chain is the same, $\Delta\omega$. What happens in presence of disorder, when this detuning is somewhat different for each of the rings? Can arbitrary unitary transformations be achieved exactly if disorder is below a certain level, or does even weak disorder prevent the realization of arbitrary transformations? Again, some examples and plots would be helpful to illustrate this experimentally relevant aspect to the readers.

3) All the examples are presented in the regime where the global phase is not fixed, according to the modulus in Eq. (6). Would it be possible to achieve arbitrary unitary transformations with a pre-defined specific global phase? This could be important if the transformation is embedded in a larger optical circuit.

Overall, the manuscript can be suitable for publication after a revision.

Reviewer #3 (Remarks to the Author):

Arbitrary linear transformations for photons in the frequency synthetic dimension
Siddharth Buddhiraju et al.

Review

The authors describe an approach for performing arbitrary linear matrix operations using optical signals. Like prior work, input signals are carried on different wavelengths. Unlike prior work, transformations directly convert between wavelengths as opposed to using multiple modes in different waveguides. This inter-wavelength coupling, termed a synthetic dimension, is achieved by breaking the time invariance of the circuit using electrooptic modulators. The processing task considered (matrix vector multiplication) is relevant, and the approach is original and innovative. The reduction in number of electrooptic elements could be highly impactful.

I have one primary reservation about the feasibility of implementing this strategy in practice. The required parameter tolerances appear to be several orders-of-magnitude tighter than is currently

possible, and this should be discussed in detail if not studied with more simulations. There are some key points that need clarification. Pending major revisions, I would recommend this manuscript for publication.

1. General questions

- a. Does the optimizer change the driving waveforms? Please clarify in text
- b. It is not entirely clear from the text how the operation is affected by the relative phases between different wavelengths. This matters because it would necessitate a particular type of comb laser. What about the phase between θ_l for different l 's?
- c. Fabricating precise κ 's poses a major challenge to this approach overall
 - + κ varies by roughly 1% - 5%, depending on the fabrication process, even for nominally identical devices on the same die. See Z. Lu, J. Jhoja et al. "Performance prediction for silicon photonics integrated circuits with layout-dependent correlated manufacturing variability," Op.Ex. 2017: <https://doi.org/10.1364/OE.25.009712> for statistics on extinction ratio
 - + Judging by the 3-5 significant figures in Supp. Mat. IV, it seems that practical device variability was not considered.
 - + I think it is important to address this aspect of how tolerant your approach will be with an inability to precisely set κ 's.
 - + You can assume that you can measure κ to high accuracy after it is fabricated. Perhaps there is some way to compensate with the waveforms
- d. How configurable is a given device? Just one transform, arbitrarily configurable transforms, or something in between?

2. Theory

- a. $\Omega_R = c/nL = c/(2 \pi n R)$
- b. Eq. S5: Is it possible to simplify by reversing the sum order? If so, I think that would help the reader see where S6 and S9 come from.
 - + $\sum_m e^{i m \Omega t} [\text{term1} + \sum_l (\text{term2})]$

3. Presenting numbers

- a. Please include the next digit when expressing numbers close to 0 or 1, for example, 0.00% => 0.003%. Expressions like ">99.99%" or "exceeding 99.99%" are fine.
- b. Preferably, use powers of 10 to indicate values close to 0 or 1. For example, 0.00003 => 3.0×10^{-5} , and 0.99994 => $1 - 6.0 \times 10^{-5}$. This is important information for quantum systems in which system fidelity is generally some high-order polynomial of device fidelity.
- c. The fidelity 79.999%: it is bizarre that there are repeated nines - one out of 10,000 chance. Could you explain why this happened (or just round it)?

4. Minor notes

- a. Fig 1b. It would be helpful to show that the inset signal is electrical and going in to the green rectangle using an arrow instead of a zoom-in symbol.
- b. Also could you put arrows in Fig 1b to show propagation direction of light
- c. Fig 4,5,6: The axes are missing labels

Reviewer 2

This is a theoretical proposal of a scheme to implement linear transformations of frequency combs. This is based on a chain of ring resonators, each driven by a modulator with specially optimized driving function. The potential to realize such arbitrary unitary and non-unitary transformation would be of interest for multiple applications in classical and quantum photonics, and thus can stimulate further experimental works. Therefore, this manuscript appears timely and suitable for a readership of Nature Communications.

Response: We thank the reviewer for his/her positive assessment of our manuscript.

The manuscript is clearly written. I have the following comments and suggestions:

1) While the intrinsic resonator loss is included in Eq. (1), it is assumed to be zero in the rest of the manuscript. In presence of such loss, would it be possible to still implement the unitary matrix transformations, up to an overall loss (such that the largest singular value is less than one)? A numerical example, e.g. in Supplementary materials, would be very useful.

Response and revision: For sufficiently small intrinsic losses, the system of Fig. (1) would be able to implement an arbitrary unitary matrix up to an overall attenuation factor, since the number of degrees of freedom in a scaled unitary matrix is the same as that of a unitary matrix. We now include an example in the Supplementary Materials (Sec. IV) depicting the fidelities achieved by a sequence of lossy resonators implementing a scaled permutation matrix. As expected, since the individual transformations implemented by the modulated resonators are not unitary in the presence of intrinsic losses, the fidelities of the overall transformations are reduced compared to the lossless case.

2) The model assumes that the frequency detuning of all the rings in a chain is the same, $\Delta\omega$. What happens in presence of disorder, when this detuning is somewhat different for each of the rings? Can arbitrary unitary transformations be achieved exactly if disorder is below a certain level, or does even weak disorder prevent the realization of arbitrary transformations? Again, some examples and plots would be helpful to illustrate this experimentally relevant aspect to the readers.

Response and revision: Yes, for a sufficiently small $\Delta\omega$ comparable to the resonance linewidths, each resonator can still implement nontrivial unitary transformations and the overall system can achieve arbitrary transformations. We now include an example in the Supplementary Materials (Sec. V) depicting the maximum achievable fidelities when $\Delta\omega/\gamma$ is chosen randomly between 0 and 10 for each of the ring resonators in the presence of a maximum modulation depth of $|\kappa_i| \leq 5\gamma$. It is seen that while the maximum achievable fidelities reduce slightly in presence of such disorder, the performance is remarkably robust to disorder.

3) All the examples are presented in the regime where the global phase is not fixed, according to the modulus in Eq. (6). Would it be possible to achieve arbitrary unitary transformations with a

pre-defined specific global phase? This could be important if the transformation is embedded in a larger optical circuit.

Response: Yes, an arbitrary unitary transformation with a pre-defined global phase can indeed be implemented. By removing the absolute value in Eq. (6), the objective function for the optimization algorithm can be redefined to capture the exact global phase. However, achieving a very high fidelity with an exact global phase may require more rings and/or modulation frequency tones to accurately match the phase at each matrix element. In order to reduce the requirement of adding more ring resonators, we instead envision achieving the transformation up to a global phase and then using a tunable delay line to add in the desired phase correction. One possible concern with using a delay line is the difference in the accumulated phase between various modulation frequency sidebands, as this phase varies linearly with frequency for a certain delay line length. We estimate this phase error to be $< 10^{-5}$ per sideband for a modulation frequency of 1 GHz, assuming a large target global phase of π at 1550 nm achieved using a ~ 350 nm delay line in lithium niobate. Therefore, the relative phase shift between sidebands is negligible and a uniform global phase can be implemented to high accuracy using a delay line.

Overall, the manuscript can be suitable for publication after a revision.

Response: We believe that our revisions have addressed the reviewer's concerns.

Reviewer 3

The authors describe an approach for performing arbitrary linear matrix operations using optical signals. Like prior work, input signals are carried on different wavelengths. Unlike prior work, transformations directly convert between wavelengths as opposed to using multiple modes in different waveguides. This inter-wavelength coupling, termed a synthetic dimension, is achieved by breaking the time invariance of the circuit using electrooptic modulators. The processing task considered (matrix vector multiplication) is relevant, and the approach is original and innovative. The reduction in number of electrooptic elements could be highly impactful.

I have one primary reservation about the feasibility of implementing this strategy in practice. The required parameter tolerances appear to be several orders-of-magnitude tighter than is currently possible, and this should be discussed in detail if not studied with more simulations. There are some key points that need clarification. Pending major revisions, I would recommend this manuscript for publication.

Response: We thank the reviewer for his/her detailed review. The concerns regarding the parameter tolerances will be addressed in the detailed response below.

1. General questions

a. Does the optimizer change the driving waveforms? Please clarify in text

Response: The optimizer changes the values of κ_l/γ , which is directly proportional to the driving waveform, in order to tune the transformation implemented by the resonators towards the target matrix. Since the driving waveform has multiple modulation tones labeled by l , the optimization does indeed end up changing the time domain waveform, or equivalently the amplitudes and phases of the frequency domain components of the waveform.

Revision: We have added a sentence at the end of the first paragraph of the Results section, reading, "Therefore, we optimize the amplitude and phase of κ_l (in units of γ) for N_r rings and N_f modulation tones per ring to implement a variety of transformations."

b. It is not entirely clear from the text how the operation is affected by the relative phases between different wavelengths. This matters because it would necessitate a particular type of comb laser. What about the phase between Θ_l for different l 's?

Response: The system we describe in the manuscript does not require a specific relative phase between the input wavelengths. The system implements a linear transformation A which acts on an input "vector" x and results in the output Ax . If relative phases between the input frequencies cause the input vector to be $x' \neq x$, then the output will be Ax' . In other words, the system of modulated ring resonators will still implement the same transformation A , since this only depends on the voltage waveform input to the modulators.

c. Fabricating precise κ 's poses a major challenge to this approach overall + κ varies by roughly 1% - 5%, depending on the fabrication process, even for nominally identical devices on the same die. See Z. Lu, J. Jhoja et al. "Performance prediction for silicon photonics integrated circuits with layout-dependent correlated manufacturing variability," Op.Ex. 2017: <https://doi.org/10.1364/OE.25.009712> for statistics on extinction ratio + Judging by the 3-5 significant figures in Supp. Mat. IV, it seems that practical device variability was not considered.

+ I think it is important to address this aspect of how tolerant your approach will be with an inability to precisely set κ 's.

+ You can assume that you can measure κ to high accuracy after it is fabricated. Perhaps there is some way to compensate with the waveforms

Response and revision: We thank the reviewer for bringing up this issue. We believe that by κ the reviewer refers to the resonator-waveguide coupling since he/she mentions the statistics of the extinction ratio. In our manuscript, κ is the frequency-dimension coupling induced by index modulation, whereas the resonator-waveguide coupling coefficient is γ .

Our approach is in fact robust to variations in γ across rings. When the ring is pumped on resonance, the transformation implemented by the ring-waveguide system is completely described by the ratios κ_l/γ , where κ_l is the modulation strength at the l^{th} modulation tone. Our optimization algorithm then prescribes the optimum values of these ratios. Since the coefficient γ can be calibrated after fabrication, the requisite modulation strengths κ_l on any ring can therefore be accurately determined by multiplying the optimum ratio prescribed by

the optimization algorithm by the γ of that ring. We now add a sentence at the end of the first paragraph in the Results section, reading “Notice that since we only optimize for the ratios κ_l/γ , our approach is robust to variations in γ during fabrication.”

Furthermore, when pumping off-resonance with a detuning $\Delta\omega$, the transformation implemented by the ring-waveguide system depends not only on κ_l/γ but also on $\Delta\omega/\gamma$. For this case, we consider in the Supplementary Material (Sec. V) a scenario where the $\Delta\omega$ of each of the rings varies randomly between 0 and 10 (in units of that ring’s γ) while the modulation strength κ_l/γ is restricted to vary between 0 and 5. We show that despite such variation, the performance is remarkably robust to disorder and very high fidelities can still be achieved.

d. How configurable is a given device? Just one transform, arbitrarily configurable transforms, or something in between?

Response: We address this question in the last paragraph on p. 5, starting “The main objective of our paper is to show...”. A system consisting of N_r rings, each with a modulator supporting N_f modulation tones, can implement an arbitrary unitary transformation of size $N \times N$ with a high fidelity provided $2N_f N_r \geq N^2$. In other words, with sufficient N_r and N_f , a given physical system can be reconfigured to implement any $N \times N$ unitary transformation by choosing the modulation amplitudes and phases as prescribed by the optimization algorithm for that target transformation. Likewise, arbitrary and reconfigurable non-unitary transformations can be achieved by first embedding them in a larger unitary transform or by using amplitude modulation, as discussed on p. 12-13 and Fig. 6.

2. Theory

a. $\Omega_R = c/nL = c/(2\pi nR)$

Response and revision: Ω_R is the FSR measured in angular frequency units (rad/s), and hence $\Omega_R = 2\pi c/nL = c/nR$. The formula the reviewer states is correct for FSR measured in frequency units (Hz). We have reworded the sentence defining Ω_R to read “...and $\Omega_R = c/nR$ is the free spectral range (FSR) of the ring in angular frequency units...”

b. Eq. S5: Is it possible to simplify by reversing the sum order? If so, I think that would help the reader see where S6 and S9 come from.

$+ \sum_m e^{im\Omega t} [\text{term1} + \sum_l (\text{term2})]$

Response and revision: We thank the reviewer for this suggestion. We have reversed the summation order and simplified Eq. S5.

3. Presenting numbers

a. Please include the next digit when expressing numbers close to 0 or 1, for example, 0.00% => 0.003%. Expressions like “>99.99%” or “exceeding 99.99%” are fine.

b. Preferably, use powers of 10 to indicate values close to 0 or 1. For example, 0.00003 => 3.0×10^{-5} , and 0.99994 => $1 - 6.0 \times 10^{-5}$. This is important information for quantum systems in which system fidelity is generally some high-order polynomial of device fidelity.

Response and revision: We thank the reviewer for this suggestion. In the revised manuscript, we now report all values in scientific notation. This change includes the fidelity maps in Figs. 4-5, which now depict *one minus* the fidelity in exponential notation to differentiate among all the values close to 1.

c. The fidelity 79.999%: it is bizarre that there are repeated nines - one out of 10,000 chance. Could you explain why this happened (or just round it)?

Response and revision: We checked the fidelity obtained in this case and the string of four nines is followed by a random sequence of numbers even with higher precision simulations. We have therefore rounded this to 80%.

4. Minor notes

a. Fig 1b. It would be helpful to show that the inset signal is electrical and going in to the green rectangle using an arrow instead of a zoom-in symbol.

Response and revision: We thank the reviewer for this suggestion. We have changed the zoom-in symbol to an arrow and replaced 'V(t)' with 'Voltage(t)' to indicate that it is an electrical signal.

b. Also could you put arrows in Fig 1b to show propagation direction of light

Response and revision: We now show the direction of propagation in the waveguide in Fig. 1b.

c. Fig 4,5,6: The axes are missing labels

Response: Figs. 4(a)-(c), 5(a)-(c) and 6(a)-(d) depict the entries of the 5 x 5 matrices and therefore do not have axes labels. On the other hand, Figs. 4(d) and 5(d) depict the performance of the inverse design algorithm as a function of the number of rings and frequencies tones, and consequently their axes have been labeled.

REVIEWER COMMENTS

Reviewer #1 (Remarks to the Author):

The revision fully addresses the previous reviewers' comments, and the manuscript is recommended for publication in the current form.

Reviewer #2 (Remarks to the Author):

I appreciate the authors' consideration of and response to my comments. Most have been addressed satisfactorily, in part by an added study of fabrication disorder (see response to reviewer 1). The work is of high quality and describes a novel and potentially impactful idea. I would suggest further clarifying one aspect of the text in plain language, and I would like to better explain one of my questions.

Since my response contains equations, it is attached as a PDF.

Arbitrary linear transformations for photons in the frequency synthetic dimension
Siddharth Buddhiraju et al.

Response to author rebuttal and revision

I appreciate the authors' consideration of and response to my comments. Most have been addressed satisfactorily, in part by an added study of fabrication disorder (see response to reviewer 1). The work is of high quality and describes a novel and potentially impactful idea. I would suggest further clarifying one aspect of the text in plain language, and I would like to better explain one of my questions.

Follow up on comment 1b:

It is correct that the phase relationship between different wavelengths is usually not relevant because they are different frequencies. What I mean is the phase relationship between the sums of different input wavelengths with their corresponding modulation phases. There is a phase matching requirement that becomes relevant in nonlinear or non-time-invariant scenarios where multiple frequencies are converted to matching frequencies. Taking a simplified example where inputs are frequencies 1 and 3, and output is frequency 2,

Phase matching condition for $1 \rightarrow 2$ coupling:

$$\omega_{2 \leftarrow 1} t + \phi_{2 \leftarrow 1} = (\omega_1 + \Omega_R)t + (\phi_1 + \theta_1)$$

Phase matching condition for $3 \rightarrow 2$ coupling:

$$\omega_{2 \leftarrow 3} t + \phi_{2 \leftarrow 3} = (\omega_3 - \Omega_R)t + (\phi_3 - \theta_3)$$

where $\omega_2 = \omega_{2 \leftarrow 1} = \omega_{2 \leftarrow 3}$ and ϕ 's denote optical phases (not to be confused with the radial position variable). An alternative statement of optical phase is

$$\phi_m = \int_0^t \omega_m(\tau) d\tau - \omega_m t$$

meaning it can only be neglected when ω_m is perfectly constant in time, which is never the case in practice.

$\phi_{2 \leftarrow 1} - \phi_{2 \leftarrow 3}$ factors strongly into the values of \mathbf{A} (thus, so does $\phi_1 - \phi_3$, supposing the θ_i 's are constant). In the extreme, if 0, the values in \mathbf{A} would have the same sign; if π , they have opposite signs. I believe the phase matching requirement makes the system sensitive to fluctuations on the optical phase variables of inputs, for example, due to relative intensity noise in the sources or even the smallest temperature fluctuation. I bring up frequency comb sources because, although their ω 's might change in time, they change together thus keeping a consistent phase relationship between wavelengths.

Ideally, but probably optionally, a term of $e^{i\phi_m}$ would be included in Eq. (S3). If the authors agree with the above, then I think this should be mentioned, explained a little bit, and at least highlighted as a topic for further work.

Suggestion on clarity:

As indicated in my comments 1a, 1c, and 1d, I found that it took some time to decipher from the equations exactly what can be controlled by modulation waveforms, both in configuring \mathbf{A} and in correcting for uncertain γ^e . I think it would be useful to some readers for this to be stated in plain language in the theory section, in particular, that κ is what is controlled (via $\delta\epsilon$), and κ/γ is what determines the transform. It is still unclear to me whether the controllability of θ_i 's is exploited for some purpose or just constant.

Reviewer #1

The revision fully addresses the previous reviewers' comments, and the manuscript is recommended for publication in the current form.

Response: We thank the reviewer for his/her positive assessment of our manuscript.

Reviewer #2

I appreciate the authors' consideration of and response to my comments. Most have been addressed satisfactorily, in part by an added study of fabrication disorder (see response to reviewer 1). The work is of high quality and describes a novel and potentially impactful idea. I would suggest further clarifying one aspect of the text in plain language, and I would like to better explain one of my questions.

Follow up on comment 1b:

It is correct that the phase relationship between different wavelengths is usually not relevant because they are different frequencies. What I mean is the phase relationship between the sums of different input wavelengths with their corresponding modulation phases. There is a phase matching requirement that becomes relevant in nonlinear or non-time-invariant scenarios where multiple frequencies are converted to matching frequencies.

...

I believe the phase matching requirement makes the system sensitive to fluctuations on the optical phase variables of inputs, for example, due to relative intensity noise in the sources or even the smallest temperature fluctuation. I bring up frequency comb sources because, although their ω 's might change in time, they change together thus keeping a consistent phase relationship between wavelengths.

Ideally, but probably optionally, a term of $e^{i\phi_m}$ would be included in Eq. (S3). If the authors agree with the above, then I think this should be mentioned, explained a little bit, and at least highlighted as a topic for further work.

Response and revision: We thank the reviewer for explaining his/her comment 1b regarding the phases of the input modes. Indeed, if one uses independently generated frequencies (e.g., separate lasers) as inputs to the system, the relative phases between them would vary slowly over time due to oscillator noise and the finite linewidth of the lasers. These lasers would need to be actively locked in phase to prevent this phase drift, something that is possible with current technology when only a few lasers are involved. As the reviewer rightly mentions, a mode-locked laser (or optical frequency comb) is a much more practical solution, since the equally spaced frequency lines of the mode-locked laser have a well-defined fixed phase relationship over time. An alternative way to generate the multi-frequency input to the system is to electro-optically modulate a single laser with a multi-tone modulation (akin to an EOM comb with a tailored spectrum).

We now include a few sentences regarding the importance of phase stability of the various frequency components at the input in the first paragraph under the Results section:

“Note that the different source frequencies' phases should not drift with respect to each other during the timescale of the transformation. To ensure such phase coherence between the different input frequency modes, the source could be a mode-locked laser or an electro-optic frequency comb with a tailored amplitude/phase spectrum to implement the input vector.”

Also, regarding Eq. (S3), the amplitude $a_m(t)$ is a complex amplitude that includes both amplitude and phase information. Thus, Eq. (S3) is a complete description of the electric field in the system and no additional phase factors are required. We have changed the relevant sentence immediately after Eq. (S3) to emphasize this point:

“...where $a_m(t)$ are the time-dependent complex amplitudes...”

Suggestion on clarity:

As indicated in my comments 1a, 1c, and 1d, I found that it took some time to decipher from the equations exactly what can be controlled by modulation waveforms, both in configuring A and in correcting for uncertain γ_e . I think it would be useful to some readers for this to be stated in plain language in the theory section, in particular, that κ is what is controlled (via $\delta\epsilon$), and κ/γ is what determines the transform. It is still unclear to me whether the controllability of θ_l 's is exploited for some purpose or just constant.

Response and revision: We thank the reviewer for this suggestion. The first paragraph under the Results section already explicitly states that the transformation is determined by the ratio κ/γ and that we optimize κ 's to implement various transformations.

In the revised manuscript, in the first paragraph under the Results section, we now further clarify that κ is determined by both $\delta\epsilon$ and θ_l , as described by Eq. (3):

“...where κ_l is controlled by the index perturbation amplitude $\delta\epsilon_l$ and phase θ_l , as described by Eq. (3).”

REVIEWERS' COMMENTS

Reviewer #2 (Remarks to the Author):

Arbitrary linear transformations for photons in the frequency synthetic dimension
Siddharth Buddhiraju et al.

Response to author response and revision #2

The authors have satisfactorily responded to my comments and made appropriate changes. I appreciate their consideration and recommend this manuscript for publication.

I would suggest adding - at the authors' discretion - a bit more about the relative phases in the manuscript. Slow fluctuations will impede the architecture's ability to function over multiple trials. In the response, active phase locking was proposed. This is indeed possible and a good idea, I think worth mentioning. Comb sources are not trivial to obtain today, so it would be beneficial to state that this other technique of slow-timescale active phase control is a possibility, rather than the architecture hinging on comb technology.

Reviewer #2

The authors have satisfactorily responded to my comments and made appropriate changes. I appreciate their consideration and recommend this manuscript for publication.

I would suggest adding - at the authors' discretion - a bit more about the relative phases in the manuscript. Slow fluctuations will impede the architecture's ability to function over multiple trials. In the response, active phase locking was proposed. This is indeed possible and a good idea, I think worth mentioning. Comb sources are not trivial to obtain today, so it would be beneficial to state that this other technique of slow-timescale active phase control is a possibility, rather than the architecture hinging on comb technology.

Response: We thank the reviewer for his/her recommendation to publish this manuscript. In accordance with his/her suggestion, we have added the following statement to the revised manuscript in the first paragraph of the Results section:

“Alternatively, active phase stabilization could be implemented to compensate for slow-timescale phase drifts.”